# Plastic Pollution: Are Bioplastics the Right Solution?

**Cristina Mastrolia** [1], **Domenico Giaquinto** [1], **Christoph Gatz** [1], **Md. Nahid Pervez** [1], **Shadi Wajih Hasan** [2], **Tiziano Zarra** [1], **Chi-Wang Li** [3], **Vincenzo Belgiorno** [1] and **Vincenzo Naddeo** [1,*]

1   Sanitary Environmental Engineering Division (SEED), Department of Civil Engineering, University of Salerno, Via Giovanni Paolo II, 84084 Fisciano, SA, Italy
2   Center for Membranes and Advanced Water Technology (CMAT), Department of Chemical Engineering, Khalifa University of Science and Technology, Abu Dhabi P.O. Box 127788, United Arab Emirates
3   Department of Water Resources and Environmental Engineering, Tamkang University, 151 Yingzhuan Road Tamsui District, New Taipei City 25137, Taiwan
*   Correspondence: vnaddeo@unisa.it; Tel.: +39-089-96-6333

**Abstract:** The adverse effects of the accumulation of plastic on our planet are no longer sustainable; plastic is a major threat to all forms of life in all environments in addition to contributing to global warming. The academic world has been focusing on registering the damages caused by plastic pollution and finding solutions to refrain from and substitute plastic and its usages, which our consumer society is so heavily dependent on. A pathway towards limiting the use of plastic comes from the European Union 2019/904 Directive for limiting the production of single-use and oxo-degradable plastics. Currently, bioplastics are one of the major alternatives in substituting fossil-based plastics, but question remain about its use. as too what extent could bioplastics be a long-term solution to plastic pollution? Is it a misconception to consider bioplastics completely harmless to the environment? This short review article aims to draw attention to the counter effects connected to the limitations and mismanagement of bioplastics through their life cycle by collecting data not published until now. A review of several cradle-to-Grave Life Cycle Assessments has been made to analyse bioplastics from production to end-of-life options. The result produced from this review article shows that bioplastics do not represent a long-term solution to plastic pollution and, on the contrary, may seem to contribute to overall environmental endangerment. The novelty of this work lies in pointing out the misconception of bioplastics' healthy effects on the environment by thoroughly analysing all environmental impacts of current production and disposal of bioplastics and by providing a more sustainable production of bioplastic through wastewater treatment plants.

**Keywords:** bioplastics; microplastics; plastic pollution; Life Cycle Assessment; circular economy

## 1. Introduction

Plastic pollution has emerged as a major issue of concern on a worldwide scale owing to the tremendous toll it takes on the environment. Although there are a variety of definitions for plastic particles, in general, those that are larger than 5 mm are referred to as macroplastics; those that are between 5 and 1 mm are known as mesoplastics; those that are between 1 mm and 0.1 μm are known as microplastics; and those that are smaller than 0.1 μm are classified as nanoplastics [1,2]. Microplastics (MPs), on the other hand, are understood to be any pieces of plastic that are smaller than 5 mm in size. Over the course of the last decade, growing scientific and social attention has been directed toward the presence of microplastics in the natural environment. The most popular ways of categorizing microplastics are as pieces, fibers, granules, pellets, and other irregular forms [3–5].

The intake of MPs may have detrimental impacts on ecosystems because of their tiny sizes, toxicity, as well as their additions and absorption of chemicals. MPs can cause mechanical harm, decrease fecundity, and alter growth rates in organisms. Primary MPs

and secondary MPs are the two broad categories used to describe microplastics. The term "primary MPs" is used to describe particles that have been manufactured on purpose keeping certain tiny sizes in mind for use in industry. Synthetic textile fibers, microbeads from cosmetics and personal care goods, and pre-production pellets used as raw materials in plastic manufacturing processes have all recently emerged as major global sources for primary MPs [6]. MPs can interact with organic contaminants via absorption and desorption [7,8] and introduce them into the food web through the ingestion of the particles by aquatic life [9]. The breakdown and weathering of major MPs items in the environment are where secondary MPs are formed. Every time a plastic bag, water bottle, or agricultural film is used, and every time a tire, marine paint, or synthetic grass is worn down, they release certain chemicals into the environment. Both marine and land-based events may serve as sources of MPs, which can then be carried by wind and surface water through the water cycle, and are eventually discharged into the oceans [10].

Recent research has shown that wastewater treatment facilities (WWTPs) may play a significant influence on the environmental release of microplastics. Human activities may directly release microbead-containing face cleansers and toothpaste into wastewater. The washing process may also release thousands of fibers from synthetic fabrics like polyester (PES) and nylon into the wastewater system [11]. Depending on the kind of treatment equipment used, the WWTP may be able to retain some of the microplastics. On the other hand, it has been demonstrated that microplastics may get through the WWTP, making their way into aquatic water sources and ultimately building up in the ecosystem [12]. These findings are concerning, since MPs may absorb harmful compounds and exert toxic effects on the skin, lungs, and digestive systems of human beings, and then these pollutants can bioaccumulate and biomagnify, presenting a threat to human health.

Bioplastics (BPLs) have been chosen as the primary substitute for petroleum-based plastics due to their organic-based origin (Table 1). However, their counter effects have been underestimated. Research has shown that bioplastics do not easily degrade in nature; in fact, their biodegradation is difficult to achieve, especially in water. It has been discovered that a complete degradation is not yet possible, but successful results have been demonstrated for the degradation of BPLs at the interface between sand and water, resulting in an almost complete degradation [13,14]. Fragments floating in the ocean are not the only negative impact of bioplastics in water. In fact, unless complete biodegradation of BPLs is reached, the remains of the disintegrated Bio-Microplastics (BMPs) might infiltrate the marine environment, causing effects only slightly less invasive than common plastics. Grievous impacts of the BMPs are recorded in sea life; the negative effects include blockage of the intestinal tract, inhibition of gastric enzyme secretion, reduced feeding necessity, decreased hormone levels, and failure to reproduce [15]. Moreover, studies have shown how the production of bioplastics has environmental impacts like petroleum-based plastics, especially during the production phase.

The aim of this short review article is to draw an overview of bioplastics, focusing on the environmental impacts (greenhouse gases potential, acidification potential, eutrophication potential, toxic potential, fossil depletion) and how to overcome their negative environmental impacts through sustainable bioplastics production by taking advantage of wastewater streams and biogas.

**Table 1.** List of bioplastics and indication of bio-based origin and biodegradability. In the table, "y" means yes, "n" means no, and "y/n" refers to both statements being valid.

| Polymer | Bio-Based | Origin | Biodegradable |
|---|---|---|---|
| Polylactic acid (PLA) | y | Starch and sugar cane | y |
| Strach blends, thermoplastic starch (TS) | y | Starch | y |
| Polyhydroxyalkanoates (PHA) | y | Bacterial fermentation | y |
| Polyhydroxybutyrate (PHB) | y | Bacterial fermentation | y |
| Polybutylene succinate (PBS) | y/n | Biological | y |
| Polyurethanes (PURs) | y/n | Chemical | y/n |
| Polucaprolactone (PCL) | n | Chemical | y |
| Polyvinyl alcohol (PVA) | n | Chemical | y |
| Polybutylene adipate terephthalate (PBAT) | n | Chemical | y |
| Polyethylene Furanoate (PEF) | y | Plants and sugar cane | n |
| Bio-polypropylene (bio-PP) | y | Sugar mill process | n |
| Polytrimethylene terephthalate (PTT) | y | Crude oil and natural gas | n |
| Bio-polyethylene terephthalate (bio-PET) | y | Sugar cane | n |
| Bio-polyethylene (bio-PE) | y | Sugar cane and wheat grain | n |
| Bio-polyamides (bio-PAs) | y | Vegetable oils | n |

## 2. Review Methodology

The present overview regarding the life cycle of bioplastics, considering both negative and positive aspects, was carried out by following the updated PRISMA 2020 statement (Preferred Reporting Items for Systematic Reviews and Meta-Analyses) [16], and the PRISMA flow diagram for this review is shown in Figure 1. The articles used for the preparation of this overview were researched through SCOPUS, SCIENCE DIRECT, GOOGLE SCHOLAR, and RESEARCHGATE databases. The keywords used for the research were "Bioplastics" OR "Bio-based Plastics" OR "Microplastics" AND "pollution" OR "degradation/biodegradation," "Life Cycle Assessment" OR "Circular Economy. The term "bioplastic" was combined with the terms "wastewater treatment plant" OR "disposal." The titles and abstracts of the searched articles were screened, and only articles directly assessing the matter of bioplastics and relevant to this review were selected. The chosen articles underwent the final reviewing stage, where the full texts were analysed for eligibility, and then the fundamental information was studied and summarised. This review represents a comprehensive analysis of bioplastics, focusing on their impact on the environment, from both production and disposal, as well as their biodegradability and recovery from biogas and wastewater treatment plants, thus making this review a novelty due to the lack of reviews elsewhere that cover all major aspects of bioplastics.

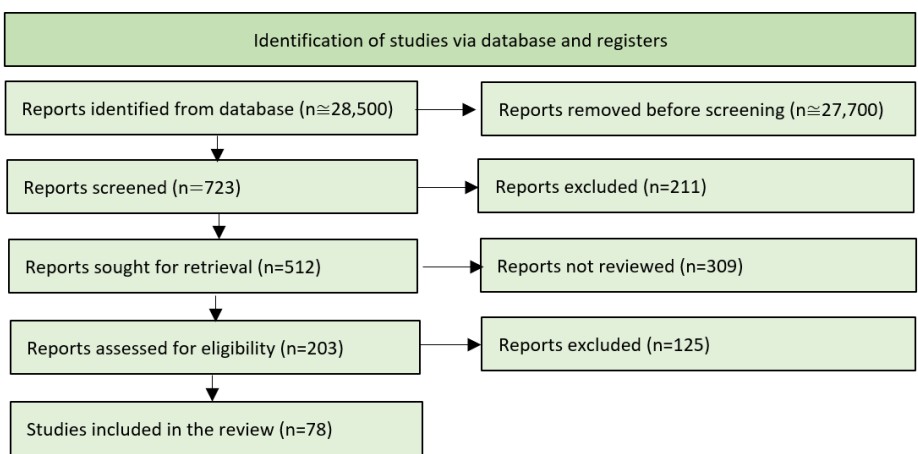

**Figure 1.** PRISMA flow diagram for this systematic review.

## 3. Plastics vs. Bioplastics

Bioplastics are the focus of research as substitutes for plastics; therefore, it is important to draw a comparison between plastics and bioplastics. Conventional plastics originate from non-biodegradable fossil-based raw materials, and thus, they are not biodegradable, which means that they cannot be decomposed by the action of microorganisms. Polymers are considered bioplastics when they meet at least one of the following criteria: (1) they originate from bio-based raw materials and/or (2) they are bio-degradable [17]. In Directive (EU) 2019/904 of the European Parliament and of the Council of 5 June 2019 on the reduction of the impact of certain plastic products on the environment, also known as the Single-use Plastics Directive, definitions for plastics and bio-degradable plastics were established. According to the Directive, plastic is defined as a material made of polymer mixed with additives or other substances which can be used to produce a final product; natural polymers that are not chemically modified are not part of this category [18]. However, biodegradable plastics are defined as a type of plastic that can be physically or biologically decomposed into biomass, carbon dioxide ($CO_2$), and water; moreover, it can be recycled by undergoing anaerobic digestion and composting [18].

A few bioplastics, such as Polybutylene Adipate Terephthalate (PBAT) and Polycaprolactone (PCL), originate from biodegradable fossil-based raw materials. However, most of the Bioplastics are produced from bio-based raw materials, either non-biodegradable like Bio-Polyethylene (Bio-PE), Polyethylene Terephthalate (Bio-PET), Polytrimethylene Terephthalate (Bio-PTT) and Bio-Polyurethane (Bio-PUR), or biodegradable such as Polylactic Acid (PLA), Polyhydroxyalkanoates (PHA), Polybutylene Succinate (PBS) and Starch Blends [19]. Even though Bio-PE, Bio-PTT, Bio-PUR, and Bio-PET do not originate from biodegradable raw materials, they are classified as BPLs due to their ability to be disposed of and recovered in a biodegradable way. A summary of biodegradable/non-biodegradable bioplastics is reported in Table 1 [17,20,21].

## 4. Biodegradability of Bioplastics

Bioplastics have been chosen as an alternative to plastic due to their ability to biodegrade in a short time, compared to the extremely long amount of time needed to biodegrade conventional plastic. The European Norm EN 13432 defines ultimate biodegradability as the "degradation mechanism characterized by the breakdown of organic chemicals by microorganisms in the presence of oxygen to carbon dioxide, water, mineral salts or other elements (mineralization), and biomass, or in the absence of oxygen to carbon dioxide, methane, mineral salts, and new biomass".

Biodegradation of biodegradable plastics can occur in an aerobic or anaerobic environment at different rates and to different extents [22], which will be discussed in the next sections.

### 4.1. Aerobic Degradation—Biodegradation in Soil and Compost

Aerobic degradation happens through a series of biochemical reactions induced by microorganisms that oxidize organic substances into water and carbon dioxide in the presence of oxygen. Biodegradation rate is measured in terms of $O_2$ consumption and $CO_2$ production [23], and it is monitored by studying the microbial assimilation index (cumulative volume of $CO_2$ compared with a blank and a positive material) and the biodegradation index (mass loss). Depending on the temperature of the aerobic environment, the biodegradation rate differs; it has been recorded that a percentage of biodegradation is approximately two times lower at low temperatures (35–37 °C) than at high temperatures (55 °C) [24,25]. Biodegradation under aerobic conditions commonly happens in soil and composting; a study by Cho et al. involving PCL-blends recorded biodegradation of 88% in 44 days [26].

Soil and compost are the preferred environments for bioplastics' degradation. The rate in the soil is affected by the soil pH, microorganisms' population, and, more importantly, humidity (optimum 50–60%) [20]. The biodegradation in a soil environment seems to be

more effective for starch-based polymers than for PHAs and PLA (one week for starch-based polymers compared to a maximum of 12 weeks for PHAs and PLA for full biodegradation).

As regards compost, most bioplastics can be effectively degraded in this environment [20]. One of the advantages of compost biodegradation is that it can take place at domestic and industrial scales; both procedures are equivalent, except for temperature levels, which are lower for the domestic scale. Certain criteria must be reached to fulfill a complete degradation in compost, making the process highly complex. To be considered compostable, a disintegration of at least 90% of the material must be broken down into fragments with a particle size of less than two mm, and at least 90% of the mass must be degraded within six months [13].

*4.2. Anaerobic Degradation—Biodegradation in Water*

Anaerobic degradation occurs in an oxygen-free environment in mesophilic (37 °C) or thermophilic (55°) conditions set in biogas plants. Without oxygen, the organic matter is transformed into water, hydrogen, carbon dioxide, methane, ammonia, and hydrogen sulphide by the metabolic interaction of different groups of microorganisms [25]. In anaerobic conditions, bioplastics reach various biodegradation results, as shown in Table 2.

**Table 2.** Bioplastics degradation rate (adapted from [25]).

| Name of the Bioplastic | Environment | Days | Degradation % | References |
|---|---|---|---|---|
| PLA | Anaerobic Digestion | 36 | 90 | [27] |
| PLA | Sludge | 60 | 85 | [28] |
| PHB | Anaerobic Digestion | 9 | 90 | [29] |
| PHB | Sludge | 9 | 90 | [30] |
| PCL | Sludge | 40–75 | 75 | [29] |
| PCL | Aquatic | 50 | 80 | [31] |

According to [31], degradation in anaerobic conditions resulted in 90% degradation of PHB in 14 days, 75% degradation of PLA in 75 days, and 80% of PCL in 50 days. More recent experiments were executed by [24], where the anaerobic degradation of four substrates was tested. The samples consisted of PE carrier bags, MaterBi® compostable bags, wine bottle corks, and cellulosic plates. The samples were pre-treated both mechanically and chemically and then mixed with inoculum and digested under mesophilic (35 ± 0.5 °C) and thermophilic conditions (55 ± 0.5 °C), following the UNI/TS 11703:2018 norm introduced in Italy. The results, based on methane emission, showed that cellulose-based plates had the highest specific methane production (75% over 10 days). Compostable bags showed 28% mass loss in 15 days and 41% after 30 days under thermophilic conditions. As for degradation under mesophilic temperature, the dry mass reduction was 23% after 15 days. The same experiment was carried out on chemically treated shredded bags, resulting in a dry mass reduction of over 78% [24].

The impact of plastic pollution is very much seen in the aquatic environment. Research has shown a fragmentation of bioplastics into BMPs, some of which can decompose in water [13]. Successful results were recorded in the eulittoral zone (BMP on the interface between sand and seawater), where an almost complete degradation was achieved [13]. However encouraging the results are that have been achieved in laboratory simulations, on-site research is still necessary. Moreover, other studies have shown the negative impact of bioplastics on aquatic environments, as they have similar properties to conventional plastics, are chemically unstable, and cannot be biologically assimilated [32].

**5. Eco-Friendliness of Bioplastics**

Biodegradability is an intrinsic characteristic of materials, which is often wrongly used as a synonym for "eco-friendly." The term "eco-friendly" is defined by the Merriam-Webster dictionary as "not environmentally harmful" [33], which is not the case for bioplastics.

To analyse the eco-friendliness of this new material, multiple studies on bioplastics' Cradle-to-Grave Life Cycle Assessments (LCA) have been compared [34]. LCA is used to determine and measure the environmental impact of Bioplastics production, use, and disposal by evaluating the complete life cycle of the material, considering extraction, production, process development, consumption, and disposal [35]. The environmental impacts analysed to study the eco-friendliness of bioplastics are [36,37]:

- Global Warming Potential (GWP): consisting in monitoring the heat absorbed by greenhouse gasses emitted during the manufacturing process caused using non-renewable energy.
- Acidification Potential (AP): consisting in monitoring the quantity of acid rain-inducing compounds, such as sulphur dioxide ($SO_2$), nitrogen oxides ($NO_x$), hydrochloric acid (HCl), ammonia ($NH_3$), and Hydrogen fluoride (HF), emitted due to the use of non-renewable energy during bioplastics production,
- Eutrophication Potential (EP): consisting in monitoring the potential to cause over-fertilization of water and soil, leading to increased growth of biomass,
- Toxic Potential (TP): consisting in monitoring the potential harm that toxins released during production and disposal can have on the environment and human health,
- Fossil Depletion (FD): consisting of the scarcity of fossils due to the use of non-renewable energy in bioplastics production.
- Non-Renewable Energy Use (NREU): consists of the use of fossil raw materials in production.

LCA has pointed out how the production process of BPLs has a significant impact on the environment. Considering the three most used BPLs (PLA, PHAs, and starch blends), a copious amount of Non-Renewable Energy Use has been recorded [38]. NREU implies the release of greenhouse gases into the atmosphere, such as carbon dioxide and methane. It cannot be overlooked that the use of non-renewable fossil-fuel energy causes the release of air pollutants such as mercury (from coal), $SO_2$, $NO_X$, and particulates that may lead to respiratory and other health problems. Furthermore, NREU also contributes to water pollution as the emission of $SO_2$ and other chemicals creates acid rain, which corrodes machinery, damages trees, and endangers animal habitats. Consequently, land integrity is likewise endangered due to the invasive extraction processes that change ecosystems as well as damage soil with the use of chemicals during the extraction process [39]. As for the impact of each BPLs, compared to conventional plastics (PP, PS, PET), studies have recorded higher NREU and GWP for PLA and PHAs than in petroleum plastic [40,41]. This is because the production of BPLs requires a lot more energy than conventional plastics. Other works have similarly found that PLA has a higher environmental impact than petroleum-based plastic in terms of AP and EP. A study by Suwanmanee et al., 2013 [40] demonstrated that PLA thermoform boxes have a higher environmental impact (GWP, AP) compared to PS plastic due to the indirect emissions from land use change, which contributes 81–91% of the impact category. As for TP, PLA has a lower impact compared to PET and PS. The cause of the greater environmental impact of PLA production is related to pellet formation, which requires higher NREU and chemicals. LCA studies on PHAs have shown similar results to PLA [34–36]; PHA demands more energy derived from fossil fuels than petrochemical plastic, resulting in more NREU and GWP. The case of the PHB composite that recorded lower GWP and NREU values than PS is different Higher values of NREU and GWP have been documented in PHA production using glucose. Moreover, in the evaluation of AP and EP, research has observed higher values in PHB than in conventional plastics. Finally, for starch-based BPLs, there are few studies that assess their environmental impact; From what is known so far, starch blends have less negative environmental impact compared to PLA and PHAs [42,43]. The abovementioned environmental impact potentials of PLA, PHAs, and PHB are summarised in Table 3.

**Table 3.** Summary of bioplastics' environmental impact potential compared to petroleum-based plastics.

| Bioplastic | NREU | GWP | AP | EP | TP | FD | References |
|---|---|---|---|---|---|---|---|
| PLA | > | = | > | > | < | = | [25,39,40,42,43] |
| PHAs | > | < | < | - | - | > | [19,32–35] |
| PHB | < | < | > | > | - | - | [19,32–35] |

>, =, <: Higher, equivalent, or lower than/to petroleum-based plastics.

## 6. Wastewater Treatment Plants in Circular Economy

Wastewater treatment plants (WWTPs) are one of the significant entry paths of microplastics into the marine environment [44,45]. Referring to Section 4 bioplastics fragmentation into bio-microplastics is also present in wastewater streams. Bio-microplastics and common microplastics affect the aquatic environment by creating a micro-habitat by forming biofilms favourable to antibiotic-resistant bacteria (ARB) and pathogens, threatening human and sea life [46].

However, WWTPs have a double-faced nature; in fact, through the application of WWTPs biological processes, bioplastics can be produced in a more sustainable way, compared to the high costs and environmental impacts consequent to the current manufacture of BPLs. To be more specific, a particular type of bioplastics (PHAs) can be produced from sludge in terms of the circular economy. The conversion of sludge into PHA can occur in Activated Sludge and Microalgae processes [47].

### 6.1. Activated Sludge

PHA is produced through the enhanced biological phosphorus removal (EBPR) system and the feast-famine mechanism [40]. In the EBPR system, it is possible to store PHA when the activated sludge process is set under the transient condition, which is a consequence of discontinuous microbial feeding and when there are variations in electron acceptor presence [48]. PHA accumulation takes place in systems characterized by the separation of an electron donor and acceptor availability or when there is a lack of substrate for microorganisms. PHA influences the ecophysiology of certain microorganisms, of Polyphosphate accumulating organisms (PAOs) and glycogen accumulating organisms (GAOs), making these two types of microorganisms the most used for PHA production. The optimal substrate used for studying the metabolisms of PAOs and GAOs is acetate. These two microbial groups can pick up acetate by consuming energy (ATP), and then it is transferred to the membrane where acetyl-CoA is activated. Consequently, acetyl-CoA plays a role in the synthesis of PHAs through the reduction of hydroxybutyryl-CoA and condensation of acetoacetyl-CoA, creating a chain of PHA [49].

As for the feast-famine mechanism, the process consists of the accumulation of PHA through the changes of the microorganism metabolisms under the selective pressure applied to the system. By doing so, microorganisms enrich the environment with strains that have PHA function, and since this mechanism is stable and strong, it makes the feast-famine process one of the most popular PHA production methods. A further study consisted of a combination of the EBPR and feast-famine processes, resulting in a more efficient PHA production [47].

### 6.2. Microalgae

The first record of the use of microalgae for wastewater treatment goes back to the 1960s [50]. Their use has been promoted due to the presence of contaminants such as organic matter (COD), phosphorus, nitrogen, iron, manganese, and others in wastewater, all of which play a role in the production of microalgae [41,51]. The production of BPLs by employing starch from microalgae is possible under high temperatures and a proper dose of glycerol [52]. However, the combination of starch with additives is unavoidable, leading to a reduction in bioplastics biodegradability if added in excess [53,54].

Cultivation of microalgae requires light, nutrients, and $CO_2$ [55,56]. Three types of reactors have been frequently used for microalgae cultivation: (i) Traditional open pond

system, (ii) Closed photobioreactor system, and (iii) hybrid systems. Open ponds are the most efficient cultivation process, achieving a nutrient removal rate of approximately 82–99% and organic matter removal of approximately 46–76% [57]. As for the closed photobioreactor system, a study employing a membrane photobioreactor (MPBR) as a microalgae cultivation process has achieved a nitrogen and phosphorus removal efficiency of approximately 82–86% [58,59]. The need to explore other cultivation systems besides open ponds comes from the limits of the latter; in fact, open ponds lead to high water evaporation, low biomass productivity, low nutrient removal, and a large footprint [47].

The accumulation of starch granules in microalgae is generated by a response under stress conditions, consisting of extreme pH and $CO_2$, nitrogen starvation, high salinity, and sulphur-deprived medium [60]. Following starch accumulation, starch is extracted by the beating method, ultrasonication, and physiochemical method [61]. It has been recorded that the highest starch recovery came from the beating method, resulting in a recovery of approximately 97%; the second best performing method is the physiochemical method, with a recovery of approximately 95% and finally, ultrasonication, resulting in a recovery of approximately 70% [61].

Wet microalgae can undergo drying-hydrolysis or lipid extraction to produce PHA after bacterial fermentation [55,62]. The production of PHAs from microalgae is achieved through: (i) acidogenic fermentation to produce volatile fatty acids (VFAs) from biodegradable organisms; (ii) selection of PHAs storing biomass in a sequencing batch reactor (SBR); and (iii) batch step to maximize PHAs accumulation in bacteria cells [63]. An example of a new processing line for microalgae cultivation and PHA production from wastewater is shown in Figure 2.

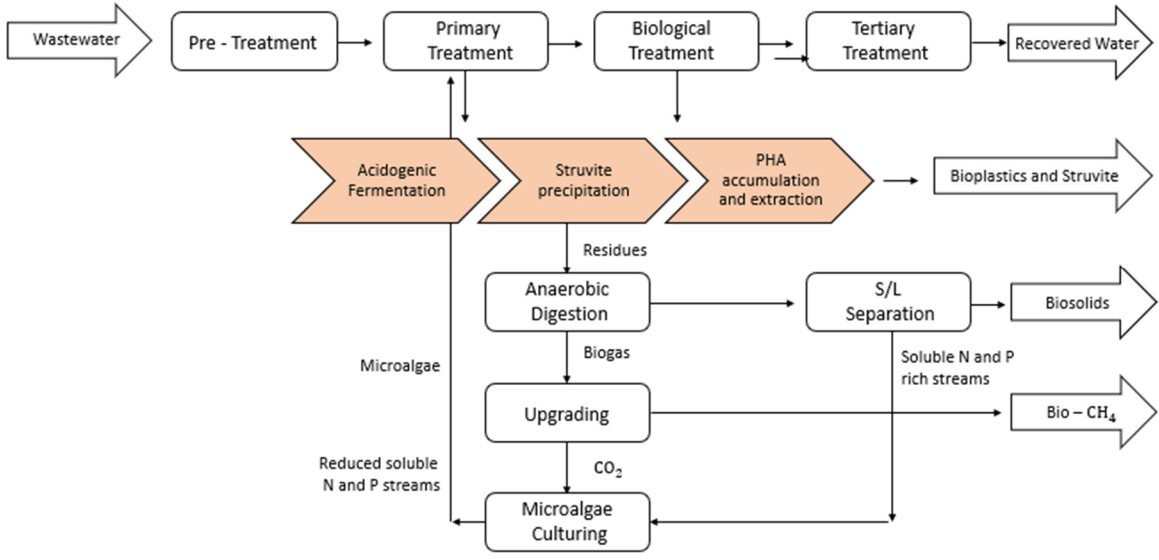

**Figure 2.** Wastewater treatment scheme for resource recovery (adapted from [63]).

Nutrients (P, N, C) are the limiting factor for bioplastics production; if the wastewater streams used as substrate are highly nutrient deficient, the produced PHA would not be stable; whereas, if the wastewater streams used are mildly nutrient deficient, the probability of producing stable PHA cultures is higher [64,65]. Other than a stable product, the advantage of using Microalgae is their independence from potable water and soil for growth, reducing costs of harvesting and cultivation. Moreover, it has been proven that co-fermentation of microalgae mixed culture liquid fraction of the digestate produced a higher amount of VAFs compared to sole microalgae fermentation [63].

Because microalgae can double their biomass within a few hours, the production time of PHAs from microalgae is reduced. Meanwhile, the low environmental impact is also a key advantage of microalgae due to the consumption of $CO_2$ and production of $O_2$ [55].

Furthermore, the employment of microalgae improves a bio-based alternative to synthetic plastics, and it takes advantage of second-generation feedstock [63].

Comparing the data between sludge production/disposal and the production of biogas and PHA from Wetterskip Fryslan's WWTP (Leeuwarden, Netherlands), showed that if about 28% of the sludge is used for PHA accumulation, around 2500 tons of bioplastic can be produced, and around 10,000 tons of $CO_2$-eq present in the WWTP can be reduced. Even with only 25% of sludge being used for the production of PHA, around 21% of carbon footprint reduction could be achieved compared to the WWTPs without PHA [63,66].

## 7. Problems Associated with Bioplastic Disposal and Possible Solutions

The final step of the LCA is disposal. Bioplastics, after production and use, can be chemically recycled or can produce renewable resources through aerobic/anaerobic digestion and composting; fragments contaminated with plastics or chemicals cannot be treated and thus end up in landfills and incineration plants [67,68].

Even though bioplastics can be used to produce new materials and energy, the disposal techniques are not impact-free. In fact, the release of pollutants, the high cost of chemicals, and the high energy consumption of chemical recycling make this end-of-life option unfeasible [69]. As for composting, it requires high temperature and pressure conditions, only achievable in laboratory conditions; thus, current composting results in low-quality products and a non-reliable market for selling. Lastly, landfills have the same drawbacks for bioplastics and conventional plastics—high costs, gas, and pollutant emissions, the latter of which result in a threat to the environment [70,71].

Miscommunication and misconception of the term biodegradable cause mismanagement of bioplastics by the public. According to the common misconception, biodegradable means naturally degradable into the environment, a convention that leads to more littering, making bioplastics as dangerous or even more dangerous than plastics in terms of negligent pollution. In addition to a misconception, the absence of policies regarding labelling and handling of bioplastic disposal makes this new material not as eco-friendly as it ought to be. To be correctly disposed of, bioplastics, currently labelled as "other plastics", must be collected separately from plastics and must be differentiated in terms of material of origin. Because of incorrect disposal and collection, bioplastics end up in landfills and incineration plants, adding up to plastics and contributing to the generation of gasses and lactate, thus worsening the environment's contamination. Lastly, high costs of disposal are a drawback for managers of disposal facilities to treat this new material [72,73].

These challenges mentioned above cannot be resolved easily. The first step toward the correct use of bioplastics is public awareness campaigns regarding the terms "biodegradable" and "eco-friendly". Furthermore, government regulations must be modified to adjust labelling and differentiated disposal of bioplastics. In addition to bureaucratic and political updates, serious modifications and improvements must be made to technology in order to produce and dispose of bioplastics in an eco-friendlier way and in terms of a circular economy [74,75]. Finally, tax reductions and funds must be established to promote the management of bioplastics in disposal facilities.

## 8. Alternative Solutions

Alternative solutions to bioplastics are currently being considered and introduced to the market. Another major substitute for plastic in single-use items is wood or paper (e.g., wooden disposable coffee stirrers and cutlery, paper cups, and plates). Although it is a natural, renewable resource, the intensive use of wood in the production of those disposables also leads to negative environmental effects, namely deforestation and soil loss. This affects the flora and fauna habitats and leads to the depletion of biotics in the ecosystem. Deforestation also has significant effects on aquatic systems. Reducing the number of forests increases sediment input to watercourses, which leads to a change in nutrient loading and water temperature, affecting both vegetation and wildlife in natural waters. As for soil loss, the removal of organic matter creates a less hospitable habitat

for soil organisms, resulting in less diversity of soil biota [76,77]. Like deforestation, the repercussions on the aquatic system are caused by the intake of sediments, resulting in the modification of the marine habitat and endangering water biota and population. It is deducible that the need for massive production to satisfy the current global consumerism, either using renewable resources or natural resources to replace plastics, may lead to significant impacts on land and water environments [78]. Therefore, bioplastics do not represent at this time an effective long-term solution to the problems addressed in this review, due to ineffective production, management, and disposal. The most sustainable way to reduce plastic pollution and the environmental impact of plastic production is to set consumption reduction targets regarding unnecessary plastic items or even prohibit their placing on the market as designated for single-use plastics in the EU Directive 2019/14.

### 9. Conclusions

The aim of this article is to highlight the problems linked to BPLs. Even though BPLs have numerous positive sides, a wide range of harmful aspects is commonly neglected. Through a comparison of various Cradle-to Grave LCA's, stages of harvesting, manufacturing, and disposal have been evaluated. Significant environmental impacts have been recorded in the harvesting and production processes, both being the major contributors to land, air, and water pollution. According to these two phases, the rising demand for BPLs materials is causing intensive farming and the usage of chemicals to improve cultivation. As for the manufacturing process, it has been outlined how the use of Non-Renewable Energy contributes to several environmental impacts such as GWP, FD, AP, EP, and finally, TP. Moreover, a particular focus has been placed on the role of WWTPs in the BPLs industry, which can play an important role in the more sustainable production of bioplastics by applying a Circular Economy approach. Examples have been displayed of how the Activated Sludge process and use of microalgae cultivated in wastewater can significantly promote an eco-friendlier production of such material. Finally, the positive and negative sides of disposal methods have been considered.

In conclusion, bioplastics do not represent, at the time being, an effective long-term solution to plastic pollution. Nevertheless, bioplastics are contributing to the overall plastic pollution due to ineffective production, management, and disposal.

**Author Contributions:** Conceptualization: C.M., D.G., C.G. and V.N.; Methodology: C.M., D.G., C.G. and M.N.P.; Formal analysis and investigation: C.M., D.G., C.G., T.Z., M.N.P. and V.N.; Writing—original draft preparation: C.M., D.G. and C.G.; Writing—review, and editing: M.N.P., S.W.H., T.Z., C.-W.L., V.B. and V.N.; Supervision: V.N. All authors have read and agreed to the published version of the manuscript.

**Funding:** This work was supported by the Sanitary Environmental Engineering Division (SEED) and grants (FARB projects) from the University of Salerno, Italy, coordinated by V. Naddeo. Grant Number: 300393FRB22NADDE.

**Institutional Review Board Statement:** Not applicable.

**Informed Consent Statement:** Not applicable.

**Data Availability Statement:** The datasets generated during the current study are available from the corresponding author on reasonable request (Vincenzo Naddeo, V.N.).

**Conflicts of Interest:** The authors declare no conflict of interest.

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
