# Peer review of "Plastic Pollution: Are Bioplastics the Right Solution?"

_water, doi:10.3390/w14223596_

Round 1

Reviewer 1 Report

A reviewer on bioplastic is indeed an interesting subject. However, it needs major changes and upgrades.

1. Abstract is without any conclusion to bioplastics

2. line 69, should also describe what bioplastics are and what they are made up of?

3. in my opinion you can not directly copy and paste entire statements of past work-line 136-141. 

4. Is not it contradictory statements? please elaborate it in your work

Line 128-Conventional plastics 128 originate from fossil-based raw materials and are not biodegradable,

Line-143: A few bioplastics, such as Polybutylene Adipate Terephthalate (PBAT) and Polycaprolactone (PCL) originate from biodegradable fossil-based raw materials. 

How can be one bioplastic then? 

5. Then those bioplastics do not meet the criteria for being called bioplastics? line 145. criteria/defination you described in line 139. 

6. In Table 1 you should provide the source of their production. second, provide any past references if you can. 

7. I think conventional bioplastic is also biodegraded but the timescale is huge (many years). The definition provided in line 158 seems not complete. 

8. MS is more of theoretical data and very few figures to make it interesting for the reader. 

9. Line 290 to 414 has many sentences with no clear ending. 

10. What is the aim of providing wastewater treatment plants in the Circular Economy

11. A total number of articles used for this review is 58 in the bibliography or reference list. I think it's not enough. 

12. Although the topic is important but I do not understand what is a special thing or novelty which makes it relevant to international readers. There are already reviews on this topic.

Author Response

A reviewer on bioplastic is indeed an interesting subject. However, it needs major changes and upgrades.

  1. Abstract is without any conclusion to bioplastics.

 - Thank you. The conclusions to bioplastics have been specified in the abstract. Please see lines 27-29.

  1. line 69, should also describe what bioplastics are and what they are made up of?

- Thank you. Please see that origin of bioplastics is added in line (74), and also in Table 1.

  1. in my opinion you can not directly copy and paste entire statements of past work-line 136-141. 
    - Thank you, The statement has been re-written. Please see lines 136-138 and 139-141.
  2. Is not it contradictory statements? please elaborate it in your work

Line 128-Conventional plastics 128 originate from fossil-based raw materials and are not biodegradable,

Line-143: A few bioplastics, such as Polybutylene Adipate Terephthalate (PBAT) and Polycaprolactone (PCL) originate from biodegradable fossil-based raw materials. 

How can be one bioplastic then? 

- Thank you. Please see that In lines 132-134, we added non-biodegradable fossil-based raw materials in order to mark the difference from biodegradable fossil-base raw material in line 147.

  1. Then those bioplastics do not meet the criteria for being called bioplastics? line 145. criteria/defination you described in line 139. 

- Thank you. Please see that line 144-146 and 149-154 has been updated.

  1. In Table 1 you should provide the source of their production. second, provide any past references if you can. 

- Thank you. Please see that Table 1 has been updated.

  1. I think conventional bioplastic is also biodegraded but the timescale is huge (many years). The definition provided in line 158 seems not complete. 

- Thank you. The definition has been modified by writing, “Bioplastics have been chosen as an alternative to plastic due to their ability to biodegrade in a short time, compared to the extremely long amount of time needed to biodegrade conventional plastic”, line 163-164.

  1. MS is more of theoretical data and very few figures to make it interesting for the reader. 
    -Thank you. Actually, in this short review, we summarize the prospect of bioplastics as an alternative solution to microplastic. We will follow your suggestions in our next paper.
  2. Line 290 to 414 has many sentences with no clear ending. 

- Thank you. We checked and corrected it.  

  1. What is the aim of providing wastewater treatment plants in the Circular Economy.

- Thank you. The aim of providing wastewater treatment plants has been specified, please see lines 298-303.

  1. A total number of articles used for this review is 58 in the bibliography or reference list. I think it's not enough. 

- Thank you, We added more references; please see.

  1. Although the topic is important but I do not understand what is a special thing or novelty which makes it relevant to international readers. There are already reviews on this topic.

- Thank you. An explanation of how this work represents a novelty has been added in the abstract. Please see lines 29-32.

Reviewer 2 Report

This review artilce summarized both positive and negative environment impacts of bioplastics by investigating the bioplastic's cradle-to-Grave life cycle Assessments. The potential negative implacts including releasing greenhouse gases and acid rain-inducing compounds, eutrophication, relasing toxins etc. Based on the analysis, tha author pointed out that bioplastic do not present an effective long-term solution to plastic pollution.  The present study represents a timely contribution to environment pollution of bioplastic. The text is well structured and written I recommend publication of this article after a minor revison.

Subtitle of 4.3 seems a part of subtitle 4.1 and subtitle of 4.4 seems a part of subtitle of 4.2. If this case, suggested combine section of 4.1 and 4.3 and sections of 4.2 and 4.4.

The discussion on the paper and wood in the conclusion section seems not relate to the bioplastic. If the author want to highlight this substitute, Suggested adding a subtitle for this  section in the text. 

Author Response

  1. This review article summarized both positive and negative environment impacts of bioplastics by investigating the bioplastic's cradle-to-Grave life cycle Assessments. The potential negative impacts including releasing greenhouse gases and acid rain-inducing compounds, eutrophication, releasing toxins etc. Based on the analysis, the author pointed out that bioplastic do not present an effective long-term solution to plastic pollution.  The present study represents a timely contribution to environment pollution of bioplastic. The text is well structured and written I recommend publication of this article after a minor revision.
  2. Subtitle of 4.3 seems a part of subtitle 4.1 and subtitle of 4.4 seems a part of subtitle of 4.2. If this case, suggested combine section of 4.1 and 4.3 and sections of 4.2 and 4.4.

- Thank you. Corrected

  1. The discussion on the paper and wood in the conclusion section seems not relate to the bioplastic. If the author want to highlight this substitute, Suggested adding a subtitle for this  section in the text. 

- Thank you. Please see that section 8 has been created according to your suggestion.

Reviewer 3 Report

I applaud the concept of this article, as well as its completion. I think it is important. Please just check the text throughout for English usage and grammar. On line 134, for example, the "l" is missing from Single-use. Also, in line 142 the word "recoverable" does not make sense; I imagine it should be "recyclable".

Author Response

I applaud the concept of this article, as well as its completion. I think it is important. Please just check the text throughout for English usage and grammar. On line 134, for example, the "l" is missing from Single-use. Also, in line 142 the word "recoverable" does not make sense; I imagine it should be "recyclable".    

  • Thank you. Corrected.

Reviewer 4 Report

This MS-Review is investigating emerging ecological issues due to use of bioplastics as a surrogate for the standard plastic material. I found this subject quite important, and it should be elaborated in the form of the review.

However, some ambiguities and shortcomings have emerged that need to be resolved. If authors adequately answer all questions, revised MS may be published.

Specific comments:

Reference no. 5 is cited through the text, though it does not contain in whole text the key words according to which the authors claim to have done this review (in lines 95 - 98). Obviously, this reference should not be in this review, according to authors’ arguing.

In Figure 1, it is stated that total of 59 studies were included in this MS-review. however, in References section there are only 58 references.

Although PHB bioplastic is mentioned in Table 2, as well as in text, there is no explanation of meaning of “PHB” abbreviation in Table 1.

Lines 265 - 283 - some of the elaborated results are not adequately showed in Table 3. Some of the higher or lower than symbols should be different, according to the results presented in the text.

Line 292 - Mentioned section 3.4 does not exist in the MS. Is it maybe 4.3?

Lines 369 - 370 - Is that sentence correct, or it should be the other way around?

Author Response

This MS-Review is investigating emerging ecological issues due to use of bioplastics as a surrogate for the standard plastic material. I found this subject quite important, and it should be elaborated in the form of the review.

However, some ambiguities and shortcomings have emerged that need to be resolved. If authors adequately answer all questions, revised MS may be published.

Specific comments:

Reference no. 5 is cited through the text, though it does not contain in whole text the key words according to which the authors claim to have done this review (in lines 95 - 98). Obviously, this reference should not be in this review, according to authors’ arguing.

  • Thank you, the keyword “Microplastics” as been added.

In Figure 1, it is stated that total of 59 studies were included in this MS-review. however, in References section there are only 58 references.

  • Thank you, the Figure has been corrected.

Although PHB bioplastic is mentioned in Table 2, as well as in text, there is no explanation of meaning of “PHB” abbreviation in Table 1.

  • Thank you, PHB has been added to Table 1

Lines 265 - 283 - some of the elaborated results are not adequately showed in Table 3. Some of the higher or lower than symbols should be different, according to the results presented in the text.

  • Thank you. The values have been corrected.

Line 292 - Mentioned section 3.4 does not exist in the MS. Is it maybe 4.3?

  • Thank you. The error has been corrected, and section 3.4 has been corrected with 4.3

Lines 369 - 370 - Is that sentence correct, or it should be the other way around?

  • Thank you. The error has been corrected; the word conception has been corrected with misconception.

Round 2

Reviewer 1 Report

This manuscript is not a great addition but still important for international readers. It can be published.